# Optimization of the Morphology of the Removal Function for Rotating Abrasive Water Jet Polishing

**DOI:** 10.3390/mi14101931

**Published:** 2023-10-14

**Authors:** Guipeng Tie, Zhiqiang Zhang, Bo Wang, Ci Song, Feng Shi, Wanli Zhang, Hailun Si

**Affiliations:** 1College of Intelligence Science and Technology, National University of Defense Technology, 109 Deya Road, Changsha 410073, China; tieguipeng@163.com (G.T.); zhiqiangzhang0425@163.com (Z.Z.); wbnudt@163.com (B.W.); shifeng@nudt.edu.cn (F.S.); zhangwanli17@nudt.edu.cn (W.Z.); sihailun@163.com (H.S.); 2Laboratory of Science and Technology on Integrated Logistics Support, College of Intelligence Science and Technology, National University of Defense Technology, 109 Deya Road, Changsha 410073, China; 3Hunan Key Laboratory of Ultra-Precision Machining Technology, Changsha 410073, China; 4Precision Optical Manufacturing and Testing Center, Shanghai Institute of Optics and Fine Mechanics, Chinese Academy of Sciences, Shanghai 201800, China

**Keywords:** rotating abrasive water jet, removal function, morphology optimization, optimization of process parameters

## Abstract

Abrasive water jet polishing has significant advantages in the manufacturing of complex optical components (such as high-slope optical component cavities) that require high-precision manufacturing. This is due to its processing process, in which the polishing tool does not make direct contact with the surface of the workpiece, and instead maintains a considerable distance. However, the removal functions of most existing abrasive water-jet polishing technologies do not possess strict symmetry, which significantly impacts the ability to correct surface figure errors. Therefore, this study implements rotating abrasive water-jet polishing based on traditional abrasive water jet processing to optimize the removal function, which turns it into a Gaussian form; thus, obtaining a type of removal function more suitable for CCOS polishing. This paper derives an empirical formula between the distance s’ from the peak removal point of the removal function to the stagnation point and the nozzle tilt angle α, based on geometric relationships and experimental results, analyzes the relationship between material removal efficiency, nozzle tilt angle, and standoff distance. Finally, this paper verifies through experiments the validity of this empirical formula under different process parameters. Therefore, this study obtains the process conditions that allow rotating abrasive water-jet polishing technology to achieve a stable Gaussian form removal function, and the appropriate process parameters to be selected in conjunction with polishing efficiency; thereby, effectively improving the removal function’s corrective ability and manufacturing efficiency. It provides theoretical support for the processing capability and process parameter selection of abrasive water-jet polishing technology, solves the problem of limited shaping capability of existing abrasive water jet tools, and significantly improves the manufacturing capability of high-end optical components.

## 1. Introduction

With the trend towards precision and complexity in modern optical systems, the demand for manufacturing complex optical components with high precision requirements, including those with high gradients and specific microstructures, is increasing. However, existing ultra-precision polishing techniques have certain limitations when applied to these types of optical components. Abrasive water jet polishing, proposed by Faehnle, is a promising polishing method in which a jet beam (usually pure water or abrasive water) is used as a carrier to envelop abrasive particles with high momentum [1,2]. Material removal is achieved through localized stress concentration and shearing between the abrasive and the workpiece surface [3,4]. This process exhibits high material processing universality, no thermal effects on the surface, no alteration of the original surface properties (mechanical and physical), weak edge effects, and stable and controllable removal functions [5,6,7]. Therefore, this technology demonstrates strong adaptability for optical components with high gradients and specific microstructures.

Existing abrasive water-jet processing techniques can be categorized into two types: jet normal incidence and jet tilted incidence. The small size of the removal function in these methods significantly improves the polishing and figuring capabilities. In the jet normal incidence process, the contour shape of the removal function exhibits either a Gaussian-like shape (for nozzle diameters smaller than 0.3 mm) or a W shape [8,9]. Decreasing the nozzle size reduces processing efficiency, while the W-shaped removal function increases the mid- to high-frequency errors on the workpiece surface, due to minimal material removal in the center. Under conditions of jet tilted incidence, although the material removal in the center is higher than at the edges, the removal function takes on a “crescent moon” shape, which lacks strict symmetry. This presents new challenges for subsequent correction of surface figure errors and allocation of residence time [10]. Additionally, for computer controlled optical surfacing (CCOS) processes, the ideal removal function should possess Gaussian-like characteristics, requiring a single-peaked distribution with maximum material removal at the center, and gradually decreasing to zero with an increasing radius [11].

To optimize the shape of the removal function, Horiuchi et al. [12] proposed setting multiple fixed-spacing processing points along the polishing scanning path. The nozzle, rotating eccentrically, resides for a certain time at each processing point to perform the polishing. The material removal at each processing position is the sum of the removal, due to the circular motion at that point and the surrounding points, resulting in a symmetric “V” shape removal function contour. Fang et al. [13] designed a vertical impact multi-position synthesis impact processing method, in which a single nozzle resides for a certain time in each jet processing area, to achieve a removal function contour with maximum central removal. Peng et al. [9] proposed a jet removal model with a slotted jet rotating around the center, finding that decreasing the nozzle diameter concentrates the energy of the jet beam, and a quasi-Gaussian removal function can be obtained under the condition of jet normal incidence. Wang et al. [14] studied the model removal characteristics at different eccentric distances using a magnetic jet device with eccentric rotation. They found that when the eccentric distance of the nozzle is 0.8L (L is the horizontal distance between the peaks and valleys of the removal function during jet normal incidence), the resulting removal function contour is closest to Gaussian characteristics. Li et al. [15] built a jet-tilted rotating polishing device for workpiece processing. The nozzle is tilted and rotates uniformly around the stagnation point of the jet beam, resulting in a quasi-Gaussian removal function with maximum central material removal and gradually decreasing removal along the edge. This removal function is beneficial for high-precision mirror finish polishing using different control algorithms. Scholars of Hong Kong Polytechnic University have made breakthroughs in the field of abrasive water jet polishing in recent years. Cao et al. [16] carried out a series of experiments to study the influencing factors of various process parameters on the removal function, and explained in detail the corresponding relationship between water pressure, standoff distance and deflection angle, and removal function. Zhang et al. [17] combined computational fluid dynamics (CFD) simulations with abrasive dynamics analysis to develop a physical model for predicting surface topography and roughness after abrasive water jet machining. The study not only provides a deeper understanding of the microscale material removal in water jet polishing, but also provides an effective method for the prediction of surface roughness in water jet polishing. Subsequently, Zhang et al. [18] studied the influence of component curvature and slope angle on the fluid flow field, through CFD analysis and modeling for free-form optical components, and accurately predicted the removal function. However, most of the existing studies modeled non-Gaussian removal functions, and even though Gaussian-like removal functions were obtained via rotation, the formation process of Gaussian-like removal functions was not explored. Therefore, this study realizes rotating abrasive water jet polishing based on conventional abrasive water jet processing to optimize the removal function, transforming it into a Gaussian shape and obtaining a removal function type more suitable for CCOS polishing. The process of forming Gaussian-shaped removal functions and the required process conditions are investigated. The influence of different nozzle tilt angles, eccentric distances, standoff distances, etc., on the shape of the removal function and processing efficiency is analyzed. This significantly enhances the figuring capability and manufacturing efficiency of the removal function, addressing the limited figuring ability of existing abrasive water jet tools and significantly improving the manufacturing capability of high-end optical components.

## 2. Experimental Setup

The abrasive water jet polishing device used during the experiments is shown in Figure 1. The experimental device consists of a polishing liquid circulation system with closed-loop jet pressure control (Figure 1d) and a five-axis CNC polishing machine tool (Figure 1a). The vertical distance between the nozzle and the workpiece is controlled by controlling the Z axis of the machine tool. As shown in Figure 1d, the jet pressure and polishing liquid temperature can also be adjusted through the control panel. Figure 1b shows a self-developed rotating jet polishing device in our laboratory. The nozzle can be rotated around the fixed axis C’ using a servo motor drive (Figure 1c). The gear reduction ratio is 2:1. The motor is a YASKAWA rotary servo motor with the model number SGM7J-01AFC6S. The motor is rated at 0.1 kW output power and 3000 rpm rated speed. The position of the nozzle’s central axis can be adjusted using a slider with a movement accuracy of 0.1 mm and an adjustment range of −40 mm to +40 mm. At the same time, the range of nozzle tilt angle can be adjusted within 45° to 90°, with a rotation accuracy of 1°, as shown in Figure 1e,f.

During the machining process, the polishing liquid temperature is controlled at 26 °C, and the jet pressure is 1.5 MPa. The CeO_2_ abrasive, with an average particle size of 3 μm, pure water, and a dispersant are used to prepare a 10% concentration of CeO_2_ polishing liquid. The dispersant used is sodium hexametaphosphate, with a mass ratio of 1:100 to water. The polishing liquid is continuously stirred using a mechanical stirrer to ensure uniform dispersion of the abrasive. The workpiece being processed is a fused quartz with a diameter of φ50 mm. The residence time at each processing point is 3 min, and the nozzle rotation speed is controlled at 70 rpm under conditions of rotational processing. In addition, explanations of symbols in the process of theoretical analysis are listed in the following table of nomenclature.

## 3. Research and Analysis of Material Removal Characteristics of Removal Function

### 3.1. Static Oblique Incidence Machining Process

The processing principle of the rotating jet machining process is shown in Figure 2. In the actual formation process of the removal function, the shape of the removal function and the material removal rate are controlled by the angle of the nozzle and the standoff distance. The data that can be directly obtained include the nozzle eccentricity distance *p_p_*, the angle of the nozzle α, and the standoff distance H.

In order to obtain a removal function with a Gaussian or Gaussian-like shape effectively, it is necessary to rotate the nozzle around the peak removal point of the removal function during the rotating jet machining process, as shown in Figure 2b. At this time, the sizes of the nozzle eccentricity distance *p_p_* and the eccentricity distance *p_r_* of the jet incidence point should satisfy the following condition:(1)$$p_{p}=\frac{H}{tan\alpha }-p_{r}$$
(2)$$p_{r}=s^{\prime }+e=s^{\prime }+0.154H\frac{cot\alpha }{sin\alpha }$$

In Equation (2), e represents the standoff point eccentricity distance, which is the horizontal distance between the standoff point and the jet incidence point [19]. The value *s*’ represents the horizontal distance between the standoff point and the peak removal point of the removal function, under the conditions of fixed oblique jet incidence. Indeed, under any standoff distance and nozzle angle conditions, if we know the corresponding value of *s*’, we can directly obtain a Gaussian or Gaussian-like shape removal function through adjusting the size of the nozzle eccentricity distance *p_p_*.

To investigate the material removal characteristics under different nozzle angles and standoff distances, the experiment analyzed the morphological features of the removal function and the material removal rate under machining conditions using nozzle angles of 45°, 50°, 55°, 60°, 65°, 70°, 75°, and 80°, and standoff distances of 8 mm, 10 mm, 12 mm, and 14 mm.

Figure 3 shows the morphological features and profile lines of the removal function under nozzle angles ranging from 45° to 80° and standoff distances ranging from 8 mm to 14 mm. It can be observed that, under the same standoff distance conditions, the profile curve of the removal function gradually changes from a single peak to a double peak as the nozzle angle increases. However, under the same angle conditions, the shape of the profile curve of the removal function does not change significantly with the increase in standoff distance. It is known that in the abrasive water jet polishing process, a stagnation point S is formed in the jet machining region, as shown in the legend of Figure 3 (α = 80°). At this point, the shear stress is at a minimum and the pressure is at a maximum, resulting in less material removal. Along the stagnation point towards the left and right sides, the shear stress shows a trend of increasing first and then decreasing [1,2], which leads to a peak in material removal at the stagnation point.

To facilitate a more intuitive understanding of the relative position relationship between the stagnation point and the peak removal point of the removal function under different nozzle angles and standoff distances, Figure 4 and Figure 5 show the change in the cross-sectional curve of the removal function with nozzle angle and standoff distance, respectively. From Figure 4, it can be observed that under the same standoff distance conditions, the distance s’ between the peak removal point and the stagnation point of the removal function decreases gradually with the increase in nozzle angle. Moreover, under different standoff distance conditions, the trend of s’ with nozzle angle α remains consistent. To further investigate the influence of standoff distance H on s’, the curve of the cross-sectional profile of the removal function with varying standoff distance is plotted in Figure 5 under conditions of a constant nozzle angle. It can be seen that under the same nozzle angle conditions, the value of s’ remains essentially unchanged with the variation in standoff distance H, indicating that the distance between the peak removal point and the stagnation point is primarily influenced by the nozzle angle.

To further investigate the specific relationship between the distance from the peak removal point to the stagnation point and the nozzle angle, a scatter plot is plotted as shown in Figure 6. The black curve in the figure represents the fitting curve of the distance s’ from the peak removal point to the stagnation point, with respect to the nozzle angle α, using a quadratic equation for curve fitting. The equation of the curve can be expressed as:(3)$$s’=3.821\times 10^{-4}\alpha^{2}-6.967\times 10^{-2}\alpha +3.862$$

Combining Equations (1)–(3), we can obtain the relationship between the nozzle eccentricity distance and the jet incidence angle and standoff distance during the generation of a Gaussian-like removal function in the machining process, as follows:(4)$$p_{p}=\frac{H}{tan\alpha }-e-s^{\prime }=H\;cot\alpha -0.154H\frac{cot\alpha }{sin\alpha }-s’\left(\alpha \right)$$
(5)$$s’\left(\alpha \right)=b_{1}\alpha^{2}+b_{2}\alpha +b_{3}$$

In the equation, *s*’(*α*) represents the relationship between s’ and the nozzle angle, expressed as a quadratic function of α in this paper. The values *b*_1_, *b*_2_, and *b*_3_ are the corresponding coefficients, where *b*_3_ represents the distance between the stagnation point and the peak removal point of the removal function under conditions of perpendicular jet incidence. This distance is typically influenced by factors such as the nozzle exit diameter. 

In addition, from Figure 4 and Figure 5, it can be observed that under the same machining time, changes in nozzle angle and standoff distance also lead to variations in material removal efficiency. Figure 7 and Figure 8 show scatter plots and line graphs illustrating the peak removal volume and volume removal volume of the material, as a function of nozzle angle and standoff distance. It can be seen that, under any standoff distance conditions, both the volume removal volume and peak removal volume decrease with increasing nozzle angle α in the range of 45° to 90°. This is due to the fact that when the jet incidence angle decreases, the horizontal component of the fluid velocity on the workpiece surface increases, leading to an increase in shear stress and, therefore, an increase in material removal. Under fixed nozzle angle conditions, the material removal volume exhibits a trend of initially increasing and then decreasing with standoff distance. At a standoff distance of H = 12 mm, the volume removal efficiency is the highest. This is because, after the polishing liquid is sprayed from the nozzle, it undergoes collision and contraction along the circulation pipeline until the nozzle exit. The probability of collision between abrasive particles is higher, and the particles cannot achieve ideal stable motion at the moment of exit. When the standoff distance is small, some abrasive particles fail to participate in the collision and shear action on the workpiece material or the impact force is insufficient, due to the collision between the frontal abrasive particles not reaching the same impact velocity as the polishing liquid. As the standoff distance increases, the motion of the abrasive particles gradually stabilizes under the driving force of the fluid, resulting in an increasing material removal volume. However, when a certain ejection distance is reached, the fluid velocity decreases due to the suction and hindering effect of air, and the impact force of the abrasive particles also decreases. Therefore, the material removal volume decreases after a certain level of standoff distance [15]. As for rotary abrasive water jet machining, since the material removal mechanism remains the same and satisfies the linear nature of material removal efficiency, the material volume removal efficiency for rotary abrasive water jet machining is consistent with the above analysis and does not require further experimental analysis.

### 3.2. Rotating Abrasive Water Jet

Through the analysis in Section 3.1, the nozzle eccentricity distance for generating a Gaussian-shaped removal function under any nozzle angle and standoff distance conditions can be obtained. In this section, further analysis is conducted on the impact of the rotational machining process on the morphology and size of the removal function.

As shown in Figure 9, the normalized curves of the cross-sectional profiles of the removal function, obtained under the conditions of nozzle angle α = 80° and standoff distance H = 8 mm, are plotted for both rotational machining (red curve) and fixed oblique machining (blue dashed line). Figure 9a shows the case where the rotation axis of the nozzle does not coincide with the peak removal point of the removal function, while Figure 9b shows the case where the rotation axis of the nozzle coincides with the peak removal point. It can be observed that under rotational machining conditions, the overall shape of the cross-sectional profile of the removal function is similar to that obtained under non-rotational machining conditions. It exhibits a double-convex peak shape on one side, along the rotation axis C’. Additionally, under rotational machining conditions, the size value D of the removal function is twice the distance from the removal function to the rotation axis on the side away from the rotation axis, as shown in Figure 9b.

Furthermore, from Figure 9b, it can be seen that the cross-sectional profile of the removal function, obtained by rotating around the peak removal point of the removal function, does not coincide with the rotated profile of the cross-sectional function obtained under fixed oblique machining conditions. The rotational machining process results in a higher material removal volume, at the same X position, compared with the fixed oblique machining process. Figure 10 illustrates the evolution of the removal function shape during the rotational machining process. Figure 10b,c show the top view and profile view of the “crescent moon” shape removal function obtained under fixed oblique machining conditions. It can be seen that under the same depth conditions, the distance from the axis line at the peak removal point to the tail of the removal function d_3_ is greater than the distance to the boundary of the cross-sectional profile of the removal function d_2_. Therefore, when the nozzle rotates around the peak removal point, more material is removed, resulting in a shape that is closer to the Gaussian shape.

We use four known sets of nozzle eccentricity distances *p_p_* and input the data of nozzle angle α and standoff distance H into Equation (4) to calculate the nozzle eccentricity distance *p_p_* that can generate Gaussian-like removal functions during the machining process, and obtain the removal function. The experimental results are shown in Figure 11. The experimental results indicate that through using the empirical formula derived in this paper to select process parameters, stable Gaussian-shaped removal functions can be obtained, effectively improving the figuring capability of this technique.

Finally, in order to verify the improvement of the figuring ability, we conducted figuring experiments to compare the figuring ability of the “crescent moon” shape removal function and the Gaussian morphology removal function obtained after our optimization. During the experiment, two 100 mm fused silica elements with similar initial surface shape were used, as in Figure 12a,d, and the spectrum of the removal function is shown in Figure 12c. The experimental parameters are shown in Table 1.

The experimental results show that in the low-frequency surface shape error (frequency range in the interval of 0–0.351 mm^−1^), the Gaussian shape removal function has a higher ability to figure the shape error than the “crescent moon” shape, as shown in Figure 12c. At the same time, the surface shape error decreases from 0.026 wave (632.8 nm) to 0.016 wave after the figuring experiment using the “crescent moon” shape removal function; however, the surface shape error decreases from 0.034 wave to 0.014 wave after the figuring experiment using the Gaussian removal function, and the figuring ability rises significantly. In addition, in the process of comparing Figure 12b,e, it is found that the “crescent moon” shaped removal function produces mid-frequency ripples on the surface after figuring, while the mid-frequency ripples are not obvious after figuring with the Gaussian shape removal function, which also corresponds to the fact that the Gaussian shape removal function in Figure 12c has better control over the mid-frequency band than the “crescent moon” shape does.

## 4. Conclusions

In this study, rotating abrasive water jet polishing was achieved based on traditional abrasive water jet machining to optimize the removal function. By exploring the process of removal function formation, the process conditions that satisfy the Gaussian-shaped formation of the removal function were obtained; thereby, effectively improving the manufacturing capability of the removal function and providing a basis for the selection of process parameters in rotating abrasive water jet machining. The main conclusions are as follows:

An empirical formula was derived for the distance s’ between the peak removal point of the removal function and the stagnation point, based on geometric relationships and experimental results. The distance s’ from the stagnation point to the peak removal point decreases gradually with increasing nozzle angle α. The standoff distance H has no effect on the distance between the stagnation point and the peak removal point of the removal function.

Through experimental analysis, it was verified that the volume removal and peak removal of the material decrease with increasing nozzle angle α in the range of 45° to 90°. However, under fixed nozzle angle conditions, the material removal presents a trend of increasing and then decreasing with standoff distance H.

By using the empirical formula obtained in this study for the distance s’ between the peak removal point of the removal function and the stagnation point, combined with the known nozzle eccentricity distance *p_p_*, nozzle angle α, and standoff distance H in the experimental process, Gaussian-shaped removal functions were obtained effectively.

In conclusion, this study obtained the process conditions for achieving stable Gaussian-shaped removal functions in rotating abrasive water jet polishing, providing theoretical support for the processing capability and selection of process parameters in abrasive water jet polishing.

## Figures and Tables

**Figure 1 micromachines-14-01931-f001:**
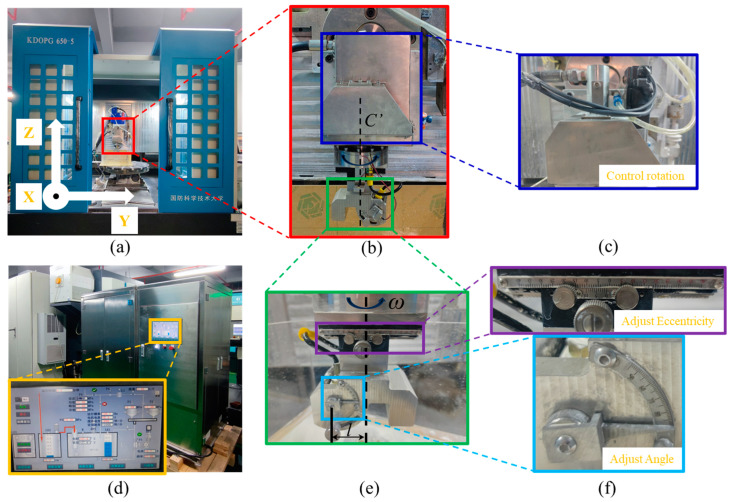
Diagram of the experimental setup: (**a**) five-axis CNC polishing machine tool; (**b**) abrasive water jet polishing head; (**c**) rotation control module for the polishing head; (**d**) circulation system and its control panel; (**e**) eccentricity and tilt control module for the nozzle; (**f**) scales for eccentricity and tilt.

**Figure 2 micromachines-14-01931-f002:**
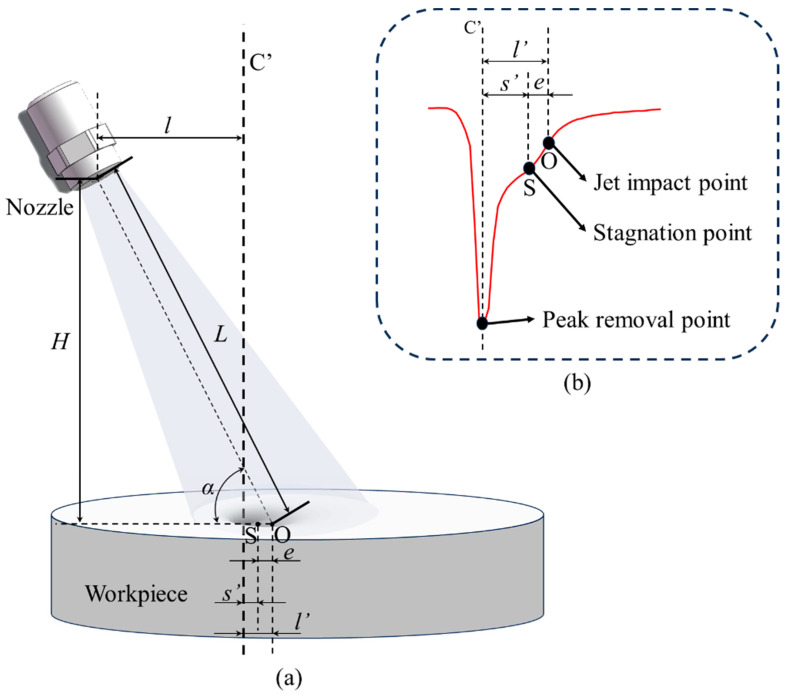
Basic schematic of the formation principle of the static oblique jet removal function. (**a**) The formation diagram of removal function. (**b**) The internal shape diagram of the remove function.

**Figure 3 micromachines-14-01931-f003:**
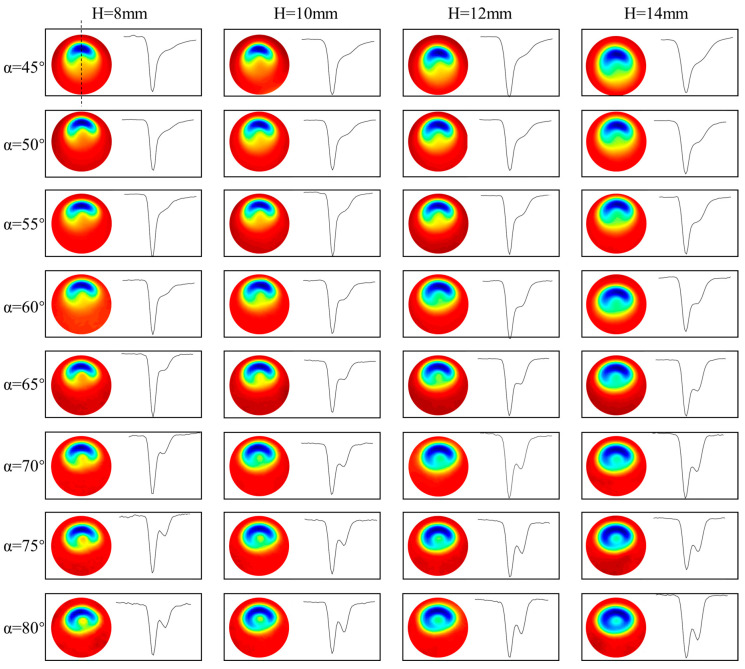
Top view and profile lines of the removal function under different nozzle angles and standoff distances.

**Figure 4 micromachines-14-01931-f004:**
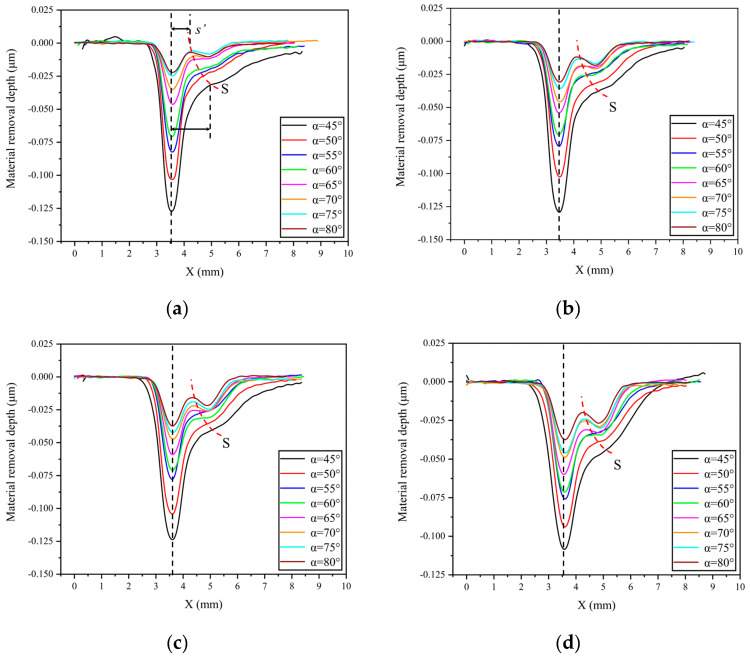
Cross−sectional curves of the removal function under constant standoff distance conditions: (**a**) H = 8 mm; (**b**) H = 10 mm; (**c**) H = 12 mm; (**d**) H = 14 mm.

**Figure 5 micromachines-14-01931-f005:**
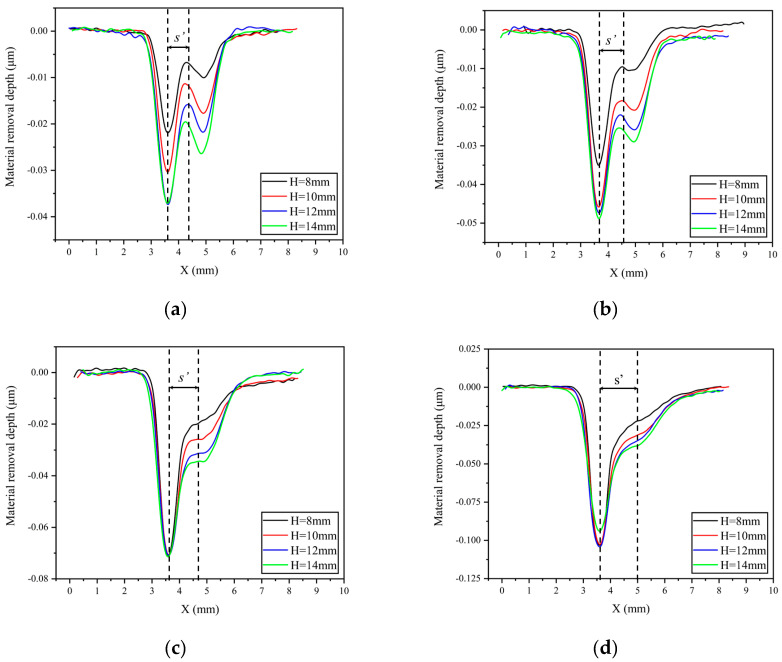
Cross−sectional curves of the removal function under constant nozzle angle conditions: (**a**) α = 80°; (**b**) α = 70°; (**c**) α = 60°; (**d**) α = 50°.

**Figure 6 micromachines-14-01931-f006:**
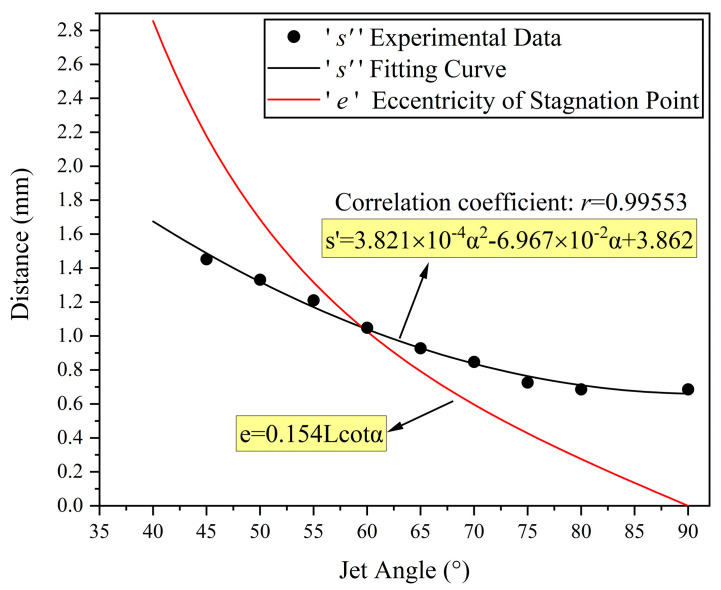
The curve of the change in distance between the peak removal point and the stagnation point of the removal function with respect to the nozzle angle.

**Figure 7 micromachines-14-01931-f007:**
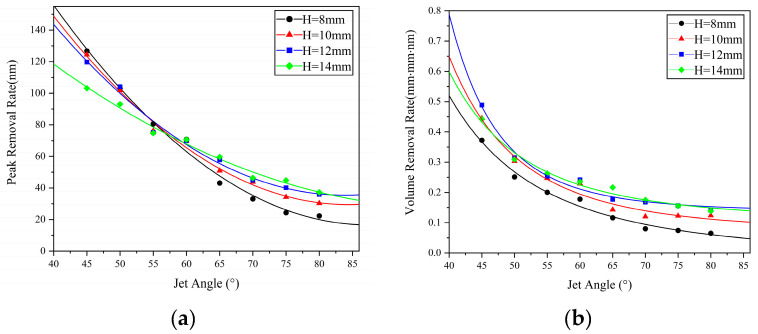
Variation in material removal rate with nozzle angle: (**a**) peak removal volume; (**b**) volume removal volume.

**Figure 8 micromachines-14-01931-f008:**
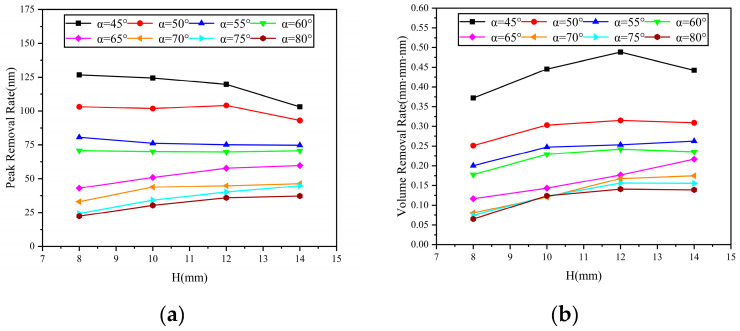
Variation in material removal rate with standoff distance: (**a**) peak removal volume; (**b**) volume removal volume.

**Figure 9 micromachines-14-01931-f009:**
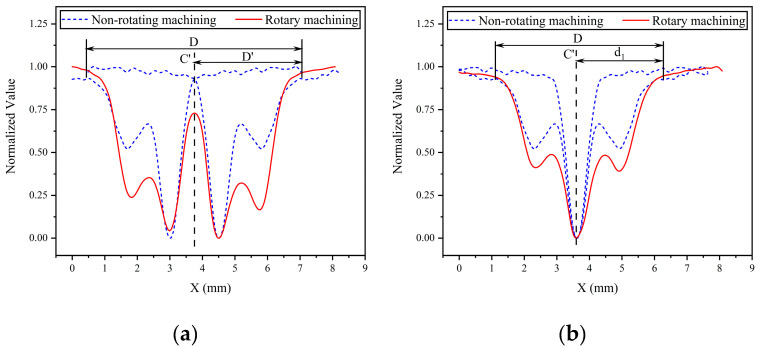
Influence of the rotational machining process on the morphology of the removal function. (**a**) the rotation axis of the nozzle does not coincide with the peak removal point of the removal function. (**b**) the rotation axis of the nozzle coincides with the peak removal point.

**Figure 10 micromachines-14-01931-f010:**
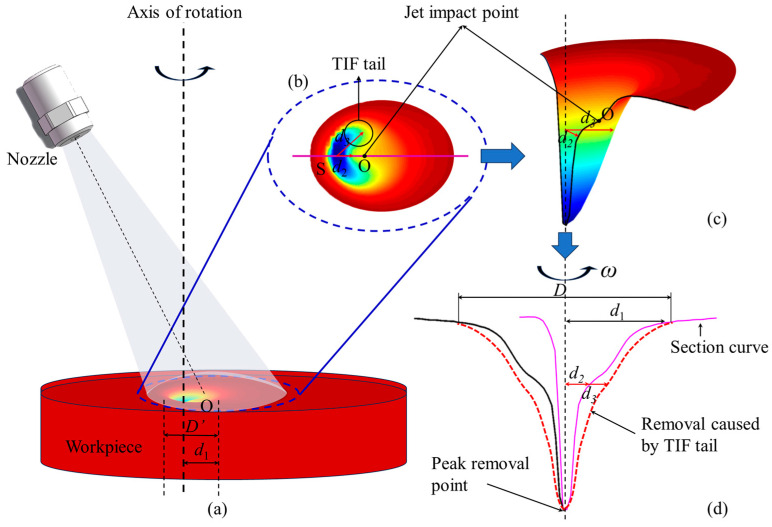
Schematic diagram of the evolution of the removal function during the rotational machining process: (**a**) abrasive water jet machining principle diagram; (**b**) top view of the removal function; (**c**) 3D profile of the removal function; (**d**) cross-sectional view of the removal function.

**Figure 11 micromachines-14-01931-f011:**
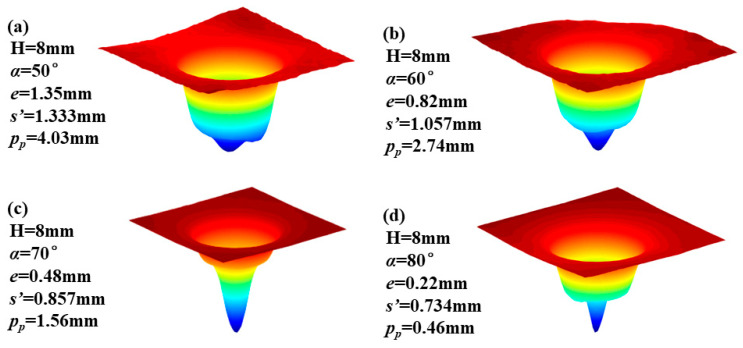
Three-dimensional morphology results of the removal function under different process parameters in rotational machining.

**Figure 12 micromachines-14-01931-f012:**
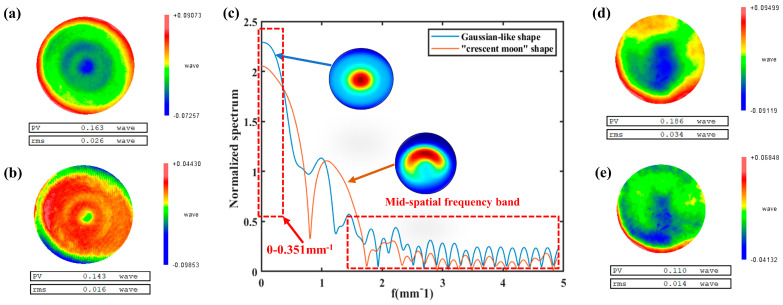
Figuring experimental results and spectrum analysis diagram: (**a**) the surface shape error before the figuring of the “crescent moon” removal function; (**b**) the surface shape error after the figuring of the “crescent moon” removal function; (**c**) remove function spectrum analysis diagram; (**d**) the surface shape error before the figuring of the Gaussian removal function; (**e**) the surface shape error after the figuring of the Gaussian removal function.

**Table 1 micromachines-14-01931-t001:** Experimental parameters.

Parameter	Value
Optical materials	Fused silica
Nozzle angle	70°
Standoff distance	8 mm
Jet pressure	1.5 Mpa
Nozzle rotation speed	69 rmp
Nozzle diameters	1 mm

## Data Availability

Not applicable.

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
