# Peer review of "Optimization of the Morphology of the Removal Function for Rotating Abrasive Water Jet Polishing"

_micromachines, 2023, doi:10.3390/mi14101931_

Round 1
Reviewer 1 Report
This study implements rotating abrasive water jet polishing based on traditional abrasive water jet processing to optimize the removal function, turning it into a Gaussian form, thus obtaining a type of removal function more suitable for CCOS polishing. The experiments in the paper validate the ideas proposed by the authors. However, some points require further elaboration by the author.
1、 Since the author has obtained the Gaussian removal function formed by the rotating water jet, why not add the modification experiment of the whole workpiece in the article.
2、 The rotating water jet device introduced by the author is not clearly visible to me in Figure 1. Whether it is similar to the bonnet polishing.
3、 Whether the machining method shown in Figure 2 is wrong, the static oblique incidence machining process should not have a rotation axis.
4、 The introduction of the development of water jet technology in the abstract is too simple, some recent developments are not introduced, such as the research on water jet technology by the Hong Kong Polytechnic University.
5、 Based on the fourth point, the references must be revised, and the citation period should be the research in recent years or the research with continuity.
Minor editing of English language required
Reviewer 2 Report
Manuscript ID: 2621500
Manuscript title: Optimization of the Morphology of the Removal Function for Rotating Abrasive Water Jet Polishing
R E V I E W
Though the paper brings an interesting subject, some questions and comments, listed below, should be considered:
1. Figure 6. Please include in the figure the value of the correlation coefficient (R) determined for the equation s’= 3.821x10-4a2-6.967x10-2a+3.862.
2. I propose to cite few recent papers (from 2019 to 2023) focused on the abrasive water jet polishing.
3. "Nomenclature" section should be included into article.
4. Figure 7. The X-axis description is missing a unit of measurement (symbol °).
In my opinion, the subject of the paper fits well within the scope of the journal and can be published in the Micromachines after corrections.

Reviewer 3 Report
The authors claim to have achieved optimization of the removal function in rotary abrasive water jet polishing, but I do not understand what they have optimized at all. The authors claim that if the relationship between a few geometrical parameters rather than the removal function in non-rotating abrasive water jet polishing and the machining conditions is known, then by rotating the workpiece with reference to this relationship, results that correlate with the non-rotating case can be obtained. Calling this "optimization" is nonsense. Furthermore, I do not believe that the axial symmetry of the red line in Figure 9(b) is relatively superior. I also do not understand how any of the results shown in Figure 11 are optimized. Therefore, the submitted paper has many problems with its claims and reliability.
Round 2
Reviewer 1 Report
It is not appropriate to separate the research results for publication. Since it is related to the paper pending review in OE Journal, the results of this paper are not suitable for publication.
appropriate
Author Response
"Please see the attachment."

Reviewer 3 Report
The author's answer seems to have cleared up my misunderstanding. It was confirmed that this paper has some significance as it suggests a method for improving the machined shape.
Author Response
Thank you for your recognition of our research
Round 3
Reviewer 1 Report
accept
fine